# Fusion approach for quantum integrable system associated with the $\mathfrak{gl}(1|1)$ Lie superalgebra

Xiaotian Xu[1,2,3], Wuxiao Wen [1], Tao Yang[1,2,3], Xin Zhang[4*], Junpeng Cao[2,4,5†]

**1** Institute of Modern Physics, Northwest University, Xi'an 710127, China
**2** Peng Huanwu Center for Fundamental Theory, Xi'an 710127, China
**3** Shaanxi Key Laboratory for Theoretical Physics Frontiers, Xi'an 710127, China
**4** Beijing National Laboratory for Condensed Matter Physics, Institute of Physics, Chinese Academy of Sciences, Beijing 100190, China
**5** School of Physical Sciences, University of Chinese Academy of Sciences, Beijing 100049, China

★ xinzhang@iphy.ac.cn , † junpengcao@iphy.ac.cn

## Abstract

In this work we obtain the exact solution of quantum integrable system associated with the Lie superalgebra $\mathfrak{gl}(1|1)$, both for periodic and for generic open boundary conditions. By means of the fusion technique we derive a closed set of operator identities among the fused transfer matrices. These identities allow us to determine the complete energy spectrum and the corresponding Bethe ansatz equations of the model. Our approach furnishes a systematic framework for studying the spectra of quantum integrable models based on Lie superalgebras, in particular when the $U(1)$ symmetry is broken. **The derivation of the Bethe states from the exact spectrum is also addressed.**

## Contents

31

32

# 1   Introduction

34  Quantum integrable models [1–3] possess significant applications in quantum field theory,
35  condensed matter physics and statistical physics, because the exact solutions of these mod-
36  els are crucial for understanding various strongly correlated effects and many-body physical
37  mechanism.

38    Quantum integrable models associated with Lie superalgebras constitute a broad subclass
39  of integrable systems [4]. Typical examples include the SU($m|n$) supersymmetric spin chains
40  [5,6], the Hubbard model [7–9], and the supersymmetric *t*-*J* model [10–12]. These models
41  have applications in a variety of fields, such as disordered electronic systems [13], critical
42  phenomena in statistical mechanics [14], and the AdS/CFT correspondence in string theory
43  [15].

44    The eigenvalue problem for this class of models can be tackled by either the coordinate
45  Bethe ansatz (CBA) or the (nested) algebraic Bethe ansatz (ABA) [16–20]. These approaches
46  hinge on the existence of a reference (or pseudo-vacuum) state. In the presence of a $U(1)$
47  symmetry, the reference state is readily constructed. However, when the $U(1)$ charge is ab-
48  sent, the construction of the reference state becomes highly non-trivial and often impossible,
49  severely limiting the applicability of the conventional Bethe ansatz techniques.

50    It has been recognized that a reference state is not indispensable for solving the spectral
51  problem. The off-diagonal Bethe ansatz (ODBA) [21] bypasses this requirement by exploiting
52  operator identities satisfied by the transfer matrix, from which Baxter's *T*-*Q* relation can be
53  constructed directly. Nevertheless, extending the ODBA to models based on Lie superalgebras
54  encounters several technical obstacles. A prominent example is the Hubbard model: in order to
55  obtain the full set of Bethe ansatz equations one still has to perform a conventional coordinate
56  Bethe ansatz or algebraic Bethe ansatz at the first nested level [18, 22], which re-introduces

57   the need for a suitable reference state.

58      Although significant progress has been made, solving integrable models associated with Lie
59   superalgebras without invoking any reference state remains an open problem. In this work
60   we address this challenge and propose a reference-state-free framework for these quantum
61   integrable systems.

62      In the present study, we focus on $\mathfrak{gl}(1|1)$, one of the most elementary Lie superalgebras.
63   In Ref. [23] Grabowski and Frahm derived the spectrum of the $\mathfrak{gl}(1|1)$ superspin chain for
64   diagonal and quasi-diagonal boundary conditions, imposing certain constraints. Their analysis
65   relied on the graded algebraic Bethe ansatz method, i.e., eigenstates were constructed by
66   acting with creation operators on a properly chosen reference state. For generic non-diagonal
67   boundary conditions, however, the construction of such a reference state becomes exceedingly
68   difficult, rendering the conventional algebraic Bethe ansatz method inapplicable.

69      The purpose of the present paper is to extend the rigorous fusion techniques introduced
70   in Refs. [24–29] to the graded case. Unlike the standard fusion procedure, we perform fusion
71   along two branches. This yields a closed set of operator identities among the fused transfer
72   matrices, from which the eigenvalue problem of the $\mathfrak{gl}(1|1)$ quantum integrable model are
73   solved exactly. With the exact spectrum in hand, we employ the separation of variables (SoV)
74   approach [30–32] to construct the Bethe state [21, 33, 34].

75      The paper is organized as follows. In Section 2, we study the integrable model associated
76   with $\mathfrak{gl}(1|1)$ under periodic boundary condition. The fusion procedure is employed to build
77   the fused transfer matrices. We obtain a closed set of operator identities that determine their
78   eigenvalues, which are parameterized by the well-known $T$-$Q$ relation. In Section 3, we extend
79   the fusion technique to the open boundary case. The eigenvalue problem of the system is
80   solved through the operator identities regarding the fused transfer matrices. Section 4 presents
81   the construction procedure for the Bethe states of the open $\mathfrak{gl}(1|1)$ integrable model. We
82   provide a conclusion in Section 5

# 2   $\mathfrak{gl}(1|1)$ **integrable model with periodic boundary**

## 2.1   **Integrability**

85   Let $V$ be a 2-dimensional $\mathbb{Z}_2$-graded linear space with a basis $\{|i\rangle | i = 1, 2\}$, where the Grass-
86   mann parities are $p(1) = 0$ and $p(2) = 1$, which endows the 2-dimensional representation of
87   the exceptional $\mathfrak{gl}(1|1)$ Lie superalgebra. The $R$-matrix $R(u) \in \text{End}(V_1 \otimes_s V_2)$ of the supersym-
88   metric $\mathfrak{gl}(1|1)$ model is [23, 35]

$$R_{1,2}(u) = \begin{pmatrix} u + \eta & & & \\ & u & \eta & \\ & \eta & u & \\ & & & u - \eta \end{pmatrix}, \tag{1}$$

89   where $u$ is the spectral parameter and $\eta$ is the crossing parameter. Here and below we adopt
90   the standard notations: for any matrix $A \in \text{End}(V \otimes_s V)$, $A_{i,j}$ is a super embedding operator of
91   $A$ in the graded tensor space, which acts as identity on the spaces except for the $i$-th and $j$-th
92   ones.

93      The $R$-matrix (1) possesses the following properties:

$$\text{regularity}: \quad R_{1,2}(0) = \eta P_{1,2}, \tag{2}$$

$$\text{unitarity}: \quad R_{1,2}(u)R_{2,1}(-u) = \rho_1(u) \times \mathbb{I}, \quad \rho_1(u) = -(u - \eta)(u + \eta), \tag{3}$$

$$\text{crossing-unitarity}: \quad R_{1,2}^{st_1}(-u)R_{2,1}^{st_1}(u) = \rho_2(u) \times \mathbb{I}, \quad \rho_2(u) = -u^2, \tag{4}$$

where $P_{1,2}$ is the super permutation operator. Here, $st_i$ is the partial super transposition $(A_{i,j}^{st_i} = A_{j,i}(-1)^{p(i)[p(i)+p(j)]})$ [36] and the super tensor product of two operators satisfies the rule $(A \otimes_s B)_{jl}^{ik} = (-1)^{[p(i)+p(j)]p(k)} A_j^i B_l^k$. The $R$-matrix (1) satisfies the graded Yang-Baxter equation (GYBE) [35, 37, 38]

$$R_{1,2}(u-v)R_{1,3}(u)R_{2,3}(v) = R_{2,3}(v)R_{1,3}(u)R_{1,2}(u-v). \tag{5}$$

We can construct the monodromy matrix $T(u)$ via the $R$-matrix (1) as

$$T_0(u) = R_{0,1}(u-\theta_1)R_{0,2}(u-\theta_2)\cdots R_{0,N}(u-\theta_N) = \begin{pmatrix} A(u) & B(u) \\ C(u) & D(u) \end{pmatrix}. \tag{6}$$

Here, $\{\theta_j | j = 1, \ldots, N\}$ are inhomogeneous parameters, the subscript 0 denotes the auxiliary space $V_0$, and the tensor product $V^{\otimes_s N}$ represents the physical (quantum) space, where $N$ is the number of lattice sites.

The monodromy matrix $T(u)$ satisfies the graded RTT relation

$$R_{1,2}(u-v)T_1(u)T_2(v) = T_2(v)T_1(u)R_{1,2}(u-v), \tag{7}$$

and can be expressed as a $2 \times 2$ matrix in the auxiliary space, whose entries are operators acting on $V^{\otimes_s N}$.

Under periodic boundary condition, the transfer matrix of the system is defined as the super trace of the monodromy matrix in the auxiliary space

$$t_p(u) = \mathrm{str}_0\{T_0(u)\} = \sum_{\alpha=1}^{2} (-1)^{p(\alpha)}[T_0(u)]_\alpha^\alpha. \tag{8}$$

With the help of the RTT relation (7), one can prove that the transfer matrices with different spectral parameters commute with each other, i.e., $[t_p(u), t_p(v)] = 0$, which guarantees the integrability of the system.

The Hamiltonian is given by the logarithmic derivative of the transfer matrix

$$\begin{aligned} H_p &= \eta \left. \frac{\partial \ln t_p(u)}{\partial u} \right|_{u=0,\{\theta_j=0\}} = \sum_{j=1}^{N} P_{j,j+1} \\ &= \sum_{j=1}^{N} \left( E_j^{11}E_{j+1}^{11} + E_j^{12}E_{j+1}^{21} + E_j^{21}E_{j+1}^{12} - E_j^{22}E_{j+1}^{22} \right), \end{aligned} \tag{9}$$

where $\{E_k^{ij}\}$ are generators of the superalgebra $\mathfrak{gl}(1|1)$, which act on the $k$-th quantum space, and the periodic boundary implies that $E_{N+1}^{ij} \equiv E_1^{ij}$. The generator $E_k^{ij}$ can be expressed in terms of the standard fermionic representation

$$E_k^{11} = 1 - n_k, \qquad E_k^{12} = c_k, \qquad E_k^{21} = c_k^\dagger, \qquad E_k^{22} = n_k,$$

where $c_j$, $c_j^\dagger$ and $n_k$ denote the fermionic annihilation, creation, and particle number operators, respectively. Therefore, the Hamiltonian (9) can be rewritten as [23]

$$H_p = \sum_{j=1}^{N} H_{j,j+1} = \sum_{j=1}^{N} \left( c_j^\dagger c_{j+1} + c_{j+1}^\dagger c_j - n_j - n_{j+1} \right) + N. \tag{10}$$

The Hamiltonian in Eq. (10) describes a model of free fermions, which can be diagonalized directly. In this paper, we solve this model in the framework of Bethe ansatz.

## 2.2 Fusion of the $R$-matrix

Fusion is a powerful and standard method for solving integrable models, particularly for those associated with high-rank Lie algebras. The $R$-matrix in integrable models degenerates into projection operators at some special points of spectral parameter $u$, which makes it possible to carry out the fused $R$-matrices and transfer matrices [24–29]. Within the conventional fusion approach, the procedure follows a single branch, as illustrated by the sequence

$$\mathfrak{t}(u) \to \mathfrak{t}^{(1)}(u) \to \mathfrak{t}^{(2)}(u) \cdots \to \mathfrak{t}^{(k)}(u).$$

The fusion procedure is considered closed when the highest-level fused transfer matrix $\mathfrak{t}^{(k)}(u)$ either becomes directly solvable [39, 40] or coincides with a transfer matrix of lower level [41, 42]. In many ordinary (non-graded) models this closure occurs after a finite number of fusion steps.

For the Lie superalgebra $\mathfrak{gl}(1|1)$ the situation is qualitatively different. The fusion of the $R$-matrix along a single branch does not yield a closed form; instead, it requires a procedure carried out along two branches, as detailed in Sections 2.2.1 and 2.2.2.

### 2.2.1 First fusion branch

**First-level fusion** At the point $u = \eta$, the $R$-matrix (1) degenerates into a 2-dimensional supersymmetric projection operator $P_{1,2}^{(+)}$

$$R_{1,2}(\eta) = 2\eta P_{1,2}^{(+)}. \tag{11}$$

Operator $P_{1,2}^{(+)}$ is defined by

$$P_{1,2}^{(+)} = \sum_{i=1}^{2} |\psi_i\rangle\langle\psi_i|, \qquad P_{1,2}^{(+)} = P_{2,1}^{(+)}, \tag{12}$$

$$|\psi_1\rangle = |1,1\rangle, \quad |\psi_2\rangle = \frac{1}{\sqrt{2}}(|1,2\rangle + |2,1\rangle), \tag{13}$$

with the parities

$$p(\psi_1) = 0, \quad p(\psi_2) = 1,$$

and projects the original 4-dimensional tensor space $V_1 \otimes_s V_2$ into a new 2-dimensional space spanned by $|\psi_1\rangle$ and $|\psi_2\rangle$. The projectors $P_{1,2}^{(+)}$ and $P_{2,1}^{(+)}$ can be obtained by exchanging two spaces $V_1$ and $V_2$, i.e., $|kl\rangle \to |lk\rangle$.

Using the projector $P_{2,1}^{(+)}$, we can construct the fused $R$-matrices

$$R_{\langle 1,2\rangle,3}(u) = (u + \tfrac{1}{2}\eta)^{-1} P_{2,1}^{(+)} R_{1,3}(u - \tfrac{1}{2}\eta) R_{2,3}(u + \tfrac{1}{2}\eta) P_{2,1}^{(+)} \equiv R_{\bar{1},3}(u), \tag{14}$$

$$R_{3,\langle 1,2\rangle}(u) = (u + \tfrac{1}{2}\eta)^{-1} P_{1,2}^{(+)} R_{3,1}(u - \tfrac{1}{2}\eta) R_{3,2}(u + \tfrac{1}{2}\eta) P_{1,2}^{(+)} \equiv R_{3,\bar{1}}(u), \tag{15}$$

where we denote the projected space by $V_{\bar{1}} = V_{\langle 1,2\rangle} = V_{\langle 2,1\rangle}$.

The fused $R$-matrix $R_{\bar{1},n}(u)$ given by (14) is a $4 \times 4$ matrix acting on the tensor space $V_{\bar{1}} \otimes_s V_n$. Its explicit form is

$$R_{\bar{1},n}(u) = \begin{pmatrix} u + \tfrac{3}{2}\eta & & & \\ & u - \tfrac{1}{2}\eta & \sqrt{2}\eta & \\ & \sqrt{2}\eta & u + \tfrac{1}{2}\eta & \\ & & & u - \tfrac{3}{2}\eta \end{pmatrix}. \tag{16}$$

143 **Second-level fusion**  At the point of $u = -\frac{3}{2}\eta$, the fused $R$-matrix defined in $R_{\bar{1},2}(u)$ (14)
144 degenerates into another projector

$$R_{\bar{1},2}(-\tfrac{3}{2}\eta) = -3\eta \mathbb{P}_{\bar{1},2}^{(-)}. \tag{17}$$

145 Here, $\mathbb{P}_{\bar{1},2}^{(-)}$ is a 2-dimensional supersymmetric projector

$$\mathbb{P}_{\bar{1},2}^{(-)} = \sum_{i=1}^{2} |\phi_i\rangle\langle\phi_i|, \tag{18}$$

146 where

$$|\phi_1\rangle = \frac{1}{\sqrt{3}}(\sqrt{2}|\psi_1\rangle \otimes_s |2\rangle - |\psi_2\rangle \otimes_s |1\rangle), \quad |\phi_2\rangle = |\psi_2\rangle \otimes_s |2\rangle. \tag{19}$$

147 The basis vectors $|\phi_1\rangle$ and $|\phi_2\rangle$ have parities

$$p(\phi_1) = 1, \quad p(\phi_2) = 0.$$

148 We see that the operator $\mathbb{P}_{\bar{1},2}^{(-)}$ projects the original 4-dimensional tensor space $V_{\bar{1}} \otimes_s V_2$ into a
149 new 2-dimensional space spanned by $|\phi_1\rangle$ and $|\phi_2\rangle$.
150      Performing the fusion procedure on $R_{\bar{1},n}(u)$ with the projector $\mathbb{P}_{\bar{1},2}^{(-)}$ yields the following
151 second-level fused $R$-matrices

$$R_{\langle\bar{1},2\rangle,3}(u) = u^{-1}\mathbb{P}_{\bar{1},2}^{(-)} R_{2,3}(u+\eta) R_{\bar{1},3}(u-\tfrac{1}{2}\eta)\mathbb{P}_{\bar{1},2}^{(-)} \equiv R_{\tilde{1},3}(u), \tag{20}$$

$$R_{3,\langle\bar{1},2\rangle}(u) = u^{-1}\mathbb{P}_{2,\bar{1}}^{(-)} R_{3,2}(u+\eta) R_{3,\bar{1}}(u-\tfrac{1}{2}\eta)\mathbb{P}_{2,\bar{1}}^{(-)} \equiv R_{3,\tilde{1}}(u). \tag{21}$$

152 Here, the projected space is denoted by $V_{\tilde{1}} = V_{\langle\bar{1},2\rangle} = V_{\langle2,\bar{1}\rangle}$. The fused $R$-matrix $R_{\tilde{1},n}(u)$ is a
153 $4 \times 4$ matrix defined in the tensor space $V_{\tilde{1}} \otimes_s V_n$ and reads

$$R_{\tilde{1},n}(u) = \begin{pmatrix} u+2\eta & & & \\ & u-\eta & -\sqrt{3}\eta & \\ & -\sqrt{3}\eta & u+\eta & \\ & & & u-2\eta \end{pmatrix}. \tag{22}$$

### 2.2.2   Second fusion branch

155 It should be noted that the $R$-matrix of the $\mathfrak{gl}(1|1)$ algebra admits another distinct fusion branch
156 beyond the one discussed above. Given the similarity of the procedure, we only present the
157 final results and detail the second fusion branch in Appendix A.
158      At the point $u = -\eta$, the $R$-matrix (1) is proportional to a projector $P_{1,2}^{(-)}$

$$R_{1,2}(-\eta) = -2\eta P_{1,2}^{(-)}. \tag{23}$$

159 By performing the fusion with the projector $P_{2,1}^{(-)}$, we obtain the first-level fused $R$-matrices

$$R_{\langle1,2\rangle',3}(u) = (u-\tfrac{1}{2}\eta)^{-1} P_{2,1}^{(-)} R_{1,3}(u+\tfrac{1}{2}\eta) R_{2,3}(u-\tfrac{1}{2}\eta)P_{2,1}^{(-)} \equiv R_{\bar{1}',3}(u), \tag{24}$$

$$R_{3,\langle1,2\rangle'}(u) = (u-\tfrac{1}{2}\eta)^{-1} P_{1,2}^{(-)} R_{3,1}(u+\tfrac{1}{2}\eta) R_{3,2}(u-\tfrac{1}{2}\eta)P_{1,2}^{(-)} \equiv R_{3,\bar{1}'}(u), \tag{25}$$

160 where the projected space is denoted as $V_{\bar{1}'} = V_{\langle1,2\rangle'} = V_{\langle2,1\rangle'}$.

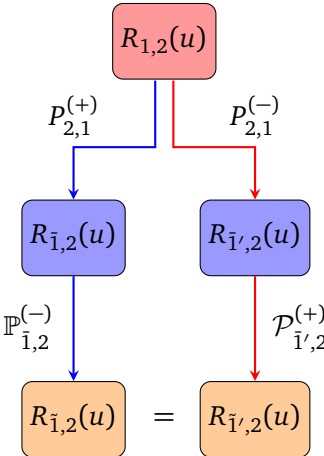

Figure 1: The fusion procedure of $R$-matrix.

At the point of $u = \frac{3}{2}\eta$, the fused matrix $R_{\bar{1}',2}(u)$ given by Eq. (24) degenerates into a projector $\mathcal{P}^{(+)}_{\bar{1}',2}$

$$R_{\bar{1}',2}(\tfrac{3}{2}\eta) = 3\eta \mathcal{P}^{(+)}_{\bar{1}',2}. \tag{26}$$

With the help of $\mathcal{P}^{(+)}_{\bar{1}',2}$, we obtain the following second-level fused $R$-matrices

$$R_{\langle \bar{1}',2\rangle,3}(u) = u^{-1} \mathcal{P}^{(+)}_{\bar{1}',2} R_{2,3}(u-\eta) R_{\bar{1}',3}(u+\tfrac{1}{2}\eta) \mathcal{P}^{(+)}_{\bar{1}',2} \equiv R_{\bar{1}',3}(u), \tag{27}$$

$$R_{3,\langle \bar{1}',2\rangle}(u) = u^{-1} \mathcal{P}^{(+)}_{2,\bar{1}'} R_{3,2}(u-\eta) R_{3,\bar{1}'}(u+\tfrac{1}{2}\eta) \mathcal{P}^{(+)}_{2,\bar{1}'} \equiv R_{3,\bar{1}'}(u), \tag{28}$$

where we denote the projected space as $V_{\bar{1}'} = V_{\langle \bar{1}',2\rangle} = V_{\langle 2,\bar{1}'\rangle}$.

### 2.2.3 Closure of the fusion

By a direct analysis, we find that $R_{\bar{1},2}(u)$ given by (20) and $R_{\bar{1}',2}(u)$ given by (27) are identical

$$R_{\bar{1},2}(u) = R_{\bar{1}',2}(u). \tag{29}$$

We perform fusion along two branches and connect the resulting fused $R$-matrices at the second fusion level. This connection thereby closes the fusion procedure, a mechanism quite different from the standard one. The fusion procedure of the $R$-matrix is briefly illustrated in Fig. 1.

## 2.3 Fused transfer matrices

The fused $R$-matrices satisfy the following graded Yang-Baxter equations

$$R_{\alpha,\beta}(u-v) R_{\alpha,\gamma}(u) R_{\beta,\gamma}(v) = R_{\beta,\gamma}(v) R_{\alpha,\gamma}(u) R_{\alpha,\beta}(u-v), \tag{30}$$

where the indices $\alpha, \beta, \gamma$ may label either the original spaces or the projected spaces.

Using the fused $R$-matrices defined in (14), (20), (24), and (27), we define the fused monodromy matrices

$$T_{\alpha}(u) = R_{\alpha,1}(u-\theta_1) R_{\alpha,2}(u-\theta_2) \cdots R_{\alpha,N}(u-\theta_N), \tag{31}$$

where the subscript $\alpha \in \{\bar{0}, \tilde{0}, \bar{0}', \tilde{0}'\}$ refers to the fused auxiliary spaces. Here, $\bar{0}$ and $\tilde{0}$ correspond to the first-level and second-level of the first fusion branch respectively; whereas $\bar{0}'$ and $\tilde{0}'$ correspond to the first-level and second-level of the second fusion branch respectively. All the fused monodromy matrices in Eq. (31) satisfy the graded RTT relations

$$R_{\alpha,\beta}(u-v)\, T_\alpha(u)\, T_\beta(v) = T_\beta(v)\, T_\alpha(u)\, R_{\alpha,\beta}(u-v). \tag{32}$$

The super traces of the fused monodromy matrices in the auxiliary spaces give the corresponding fused transfer matrices

$$\begin{aligned}
t_p^{(1)}(u) &= \mathrm{str}_{\bar{0}}\{T_{\bar{0}}(u)\}, & t_p^{(2)}(u) &= \mathrm{str}_{\bar{0}'}\{T_{\bar{0}'}(u)\}, \\
\tilde{t}_p^{(1)}(u) &= \mathrm{str}_{\tilde{0}}\{T_{\tilde{0}}(u)\}, & \tilde{t}_p^{(2)}(u) &= \mathrm{str}_{\tilde{0}'}\{T_{\tilde{0}'}(u)\}.
\end{aligned} \tag{33}$$

From Eq. (29), we conclude that the fused transfer matrices $\tilde{t}_p^{(1)}(u)$ and $\tilde{t}_p^{(2)}(u)$ are identical, we therefore denote them collectively as $\tilde{t}_p(u)$:

$$\tilde{t}_p(u) = \tilde{t}_p^{(1)}(u) = \tilde{t}_p^{(2)}(u). \tag{34}$$

The graded RTT relations in (32) imply that the transfer matrices $t_p(u)$, $t_p^{(1)}(u)$, $t_p^{(2)}(u)$ and $\tilde{t}_p(u)$ commute with each other, namely,

$$\begin{aligned}
[t_p(u), t_p^{(1)}(v)] &= [t_p(u), t_p^{(2)}(v)] = [t_p^{(1)}(u), t_p^{(2)}(v)] = 0, \\
[\tilde{t}_p(u), t_p(v)] &= [\tilde{t}_p(u), t_p^{(1)}(v)] = [\tilde{t}_p(u), t_p^{(2)}(v)] = 0.
\end{aligned} \tag{35}$$

## 2.4 Operator identities

The definitions of the fused $R$-matrices in (14), (20), (24), and (27) directly yield the following relations for the fused monodromy matrices

$$\begin{aligned}
P_{2,1}^{(+)} T_1(u) T_2(u+\eta) P_{2,1}^{(+)} &= a(u+\eta) T_{\bar{1}}(u+\tfrac{1}{2}\eta), \\
P_{2,1}^{(-)} T_1(u) T_2(u-\eta) P_{2,1}^{(-)} &= a(u-\eta) T_{\bar{1}'}(u-\tfrac{1}{2}\eta), \\
\mathbb{P}_{\bar{1},2}^{(-)} T_2(u+\eta) T_{\bar{1}}(u-\tfrac{1}{2}\eta) \mathbb{P}_{\bar{1},2}^{(-)} &= a(u) T_{\tilde{1}}(u), \\
\mathcal{P}_{\bar{1}',2}^{(+)} T_2(u-\eta) T_{\bar{1}'}(u+\tfrac{1}{2}\eta) \mathcal{P}_{\bar{1}',2}^{(+)} &= a(u) T_{\tilde{1}'}(u),
\end{aligned} \tag{36}$$

where

$$a(u) = \prod_{j=1}^N (u-\theta_j). \tag{37}$$

From the graded RTT relations (32) at specific points, together with the properties of the projectors, we derive

$$\begin{aligned}
T_1(\theta_j) T_2(\theta_j+\eta) &= P_{2,1}^{(+)} T_1(\theta_j) T_2(\theta_j+\eta), \\
T_1(\theta_j) T_2(\theta_j-\eta) &= P_{2,1}^{(-)} T_1(\theta_j) T_2(\theta_j-\eta), \\
T_2(\theta_j) T_{\bar{1}}(\theta_j-\tfrac{3}{2}\eta) &= \mathbb{P}_{\bar{1},2}^{(-)} T_2(\theta_j) T_{\bar{1}}(\theta_j-\tfrac{3}{2}\eta), \\
T_2(\theta_j) T_{\bar{1}'}(\theta_j+\tfrac{3}{2}\eta) &= \mathcal{P}_{\bar{1}',2}^{(+)} T_2(\theta_j) T_{\bar{1}'}(\theta_j+\tfrac{3}{2}\eta),
\end{aligned} \tag{38}$$

where $j = 1, \ldots, N$. Taking the super trace of Eq. (36) over the auxiliary space and using Eq. (38), we obtain the operator product identities

$$
\begin{aligned}
t_p(\theta_j)t_p(\theta_j + \eta) &= a(\theta_j + \eta)t_p^{(1)}(\theta_j + \tfrac{1}{2}\eta), \\
t_p(\theta_j - \eta)t_p(\theta_j) &= a(\theta_j - \eta)t_p^{(2)}(\theta_j - \tfrac{1}{2}\eta), \\
t_p^{(1)}(\theta_j - \tfrac{3}{2}\eta)t_p(\theta_j) &= a(\theta_j - \eta)\tilde{t}_p(\theta_j - \eta), \\
t_p^{(2)}(\theta_j + \tfrac{3}{2}\eta)t_p(\theta_j) &= a(\theta_j + \eta)\tilde{t}_p(\theta_j + \eta),
\end{aligned}
\tag{39}
$$

with $j = 1, \ldots, N$.

Figure 2 shows a schematic of the transfer matrix fusion. Unlike the conventional approach, the procedure follows two fusion branches:

$$
(1): t_p(u) \to t_p^{(1)}(u) \to \tilde{t}_p^{(1)}(u), \quad (2): t_p(u) \to t_p^{(2)}(u) \to \tilde{t}_p^{(2)}(u).
\tag{40}
$$

The fusion procedure is closed by the identity $\tilde{t}_p^{(1)}(u) = \tilde{t}_p^{(2)}(u)$. This suggests a novel strategy for solving integrable models associated with Lie superalgebra: building multiple fusion branches and connecting them to achieve a closed system.

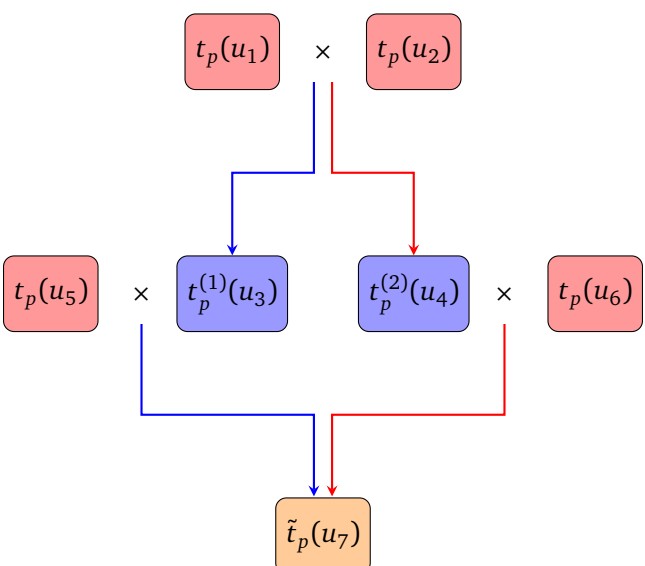

Figure 2: Schematic diagram of the transfer matrix fusion procedure. The blue and red lines represent the fist and second fusion branches respectively. The spectral parameter $u_j$ must be set to a specific value at each step, as shown in Eq. (39).

## 2.5  *T-Q* **relation**

Let $\Lambda_p(u)$, $\Lambda_p^{(1)}(u)$, $\Lambda_p^{(2)}(u)$ and $\tilde{\Lambda}_p(u)$ denote the eigenvalues of the transfer matrices $t_p(u)$, $t_p^{(1)}(u)$, $t_p^{(2)}(u)$ and $\tilde{t}_p(u)$, respectively. As the fused transfer matrices mutually commute, the operator product identities in (39) directly lead to the following functional relations

$$
\begin{aligned}
\Lambda_p(\theta_j)\Lambda_p(\theta_j + \eta) &= a(\theta_j + \eta)\Lambda_p^{(1)}(\theta_j + \tfrac{1}{2}\eta), \\
\Lambda_p(\theta_j - \eta)\Lambda_p(\theta_j) &= a(\theta_j - \eta)\Lambda_p^{(2)}(\theta_j - \tfrac{1}{2}\eta), \\
\Lambda_p^{(1)}(\theta_j - \tfrac{3}{2}\eta)\Lambda_p(\theta_j) &= a(\theta_j - \eta)\tilde{\Lambda}_p(\theta_j - \eta), \\
\Lambda_p^{(2)}(\theta_j + \tfrac{3}{2}\eta)\Lambda_p(\theta_j) &= a(\theta_j + \eta)\tilde{\Lambda}_p(\theta_j + \eta),
\end{aligned}
\tag{41}
$$

where $j = 1, \ldots, N$. Since $\Lambda_p(u)$, $\Lambda_p^{(1)}(u)$, $\Lambda_p^{(2)}(u)$, and $\tilde{\Lambda}_p(u)$ are degree-$(N-1)$ polynomials in $u$, the $4N$ constraints in Eq. (41) completely determine these functions.

We can parameterize the eigenvalues $\Lambda_p(u)$, $\Lambda_p^{(1)}(u)$, $\Lambda_p^{(2)}(u)$ and $\tilde{\Lambda}_p(u)$ in terms of the following $T$-$Q$ relations

$$
\begin{aligned}
\Lambda_p(u) &= [a(u) - a(u - \eta)]\frac{Q(u + \eta)}{Q(u)}, \\
\Lambda_p^{(1)}(u) &= [a(u - \tfrac{1}{2}\eta) - a(u - \tfrac{3}{2}\eta)]\frac{Q(u + \tfrac{3}{2}\eta)}{Q(u - \tfrac{1}{2}\eta)}, \\
\Lambda_p^{(2)}(u) &= [a(u - \tfrac{3}{2}\eta) - a(u - \tfrac{1}{2}\eta)]\frac{Q(u + \tfrac{3}{2}\eta)}{Q(u - \tfrac{1}{2}\eta)}, \\
\tilde{\Lambda}_p(u) &= [a(u - 2\eta) - a(u - \eta)]\frac{Q(u + 2\eta)}{Q(u - \eta)},
\end{aligned}
\tag{42}
$$

where

$$
Q(u) = \prod_{k=1}^{M}(u - \mu_k),
\tag{43}
$$

and $M$ is the number of Bethe roots $\{\mu_k\}$ and ranges from 0 to $N$. The analyticity of $\Lambda_p(u)$, $\Lambda_p^{(1)}(u)$, $\Lambda_p^{(2)}(u)$ and $\tilde{\Lambda}_p(u)$ requires that the Bethe roots $\{\mu_k\}$ must satisfy the Bethe ansatz equations (BAEs)

$$
\prod_{j=1}^{N}\frac{\mu_k - \theta_j - \eta}{\mu_k - \theta_j} = 1, \quad k = 1, \ldots, M.
\tag{44}
$$

The eigenvalue of the Hamiltonian (10) can be given by the Bethe roots as follows

$$
E_p = \eta\, \frac{\partial \ln \Lambda_p(u)}{\partial u}\bigg|_{u=0,\{\theta_j=0\}} = \sum_{k=1}^{M}\frac{\eta^2}{(\eta - \mu_k)\mu_k} - N.
\tag{45}
$$

Numerical results for the $N = 3$ and $N = 4$ cases are presented in Tables 1 and 2 respectively. It can be seen that the eigenvalue $E_p$ derived from the Bethe roots coincides with that from the direct diagonalization of the Hamiltonian (10).

Table 1: Numeric results of Bethe roots $\{\mu_k\}$ and eigenvalues of the Hamiltonian (10). Here, $N = 3$, $\eta = 1$ and $\{\theta_j = 0\}$.

| $\mu_1$ | $\mu_2$ | $\mu_3$ | $E_p$ |
|---|---|---|---|
| – | – | – | $-3$ |
| $\infty$ | – | – | $-3$ |
| $\frac{3 - i\sqrt{3}}{6}$ | – | – | $0$ |
| $\frac{3 + i\sqrt{3}}{6}$ | – | – | $0$ |
| $\frac{3 - i\sqrt{3}}{6}$ | $\infty$ | – | $0$ |
| $\frac{3 + i\sqrt{3}}{6}$ | $\infty$ | – | $0$ |
| $\frac{3 + i\sqrt{3}}{6}$ | $\frac{3 - i\sqrt{3}}{6}$ | – | $3$ |
| $\frac{3 + i\sqrt{3}}{6}$ | $\frac{3 - i\sqrt{3}}{6}$ | $\infty$ | $3$ |

Table 2: Numeric results of Bethe roots $\{\mu_k\}$ and eigenvalues of the Hamiltonian (10). Here, $N = 4$, $\eta = 1$ and $\{\theta_j = 0\}$.

| $\mu_1$ | $\mu_2$ | $\mu_3$ | $\mu_4$ | $E_p$ | $\mu_1$ | $\mu_2$ | $\mu_3$ | $\mu_4$ | $E_p$ |
|---------|---------|---------|---------|-------|---------|---------|---------|---------|-------|
| – | – | – | – | $-4$ | $\frac{1+i}{2}$ | $\frac{1-i}{2}$ | – | – | $0$ |
| $\infty$ | – | – | – | $-4$ | $\frac{1+i}{2}$ | $\frac{1}{2}$ | – | – | $2$ |
| $\frac{1+i}{2}$ | – | – | – | $-2$ | $\frac{1-i}{2}$ | $\frac{1}{2}$ | – | – | $2$ |
| $\frac{1-i}{2}$ | – | – | – | $-2$ | $\infty$ | $\frac{1+i}{2}$ | $\frac{1-i}{2}$ | – | $0$ |
| $\frac{1}{2}$ | – | – | – | $0$ | $\infty$ | $\frac{1-i}{2}$ | $\frac{1}{2}$ | – | $2$ |
| $\infty$ | $\frac{1+i}{2}$ | – | – | $-2$ | $\infty$ | $\frac{1+i}{2}$ | $\frac{1}{2}$ | – | $2$ |
| $\infty$ | $\frac{1-i}{2}$ | – | – | $-2$ | $\frac{1+i}{2}$ | $\frac{1-i}{2}$ | $\frac{1}{2}$ | – | $4$ |
| $\infty$ | $\frac{1}{2}$ | – | – | $0$ | $\infty$ | $\frac{1+i}{2}$ | $\frac{1-i}{2}$ | $\frac{1}{2}$ | $4$ |

Under periodic boundary conditions, the $\mathfrak{gl}(1|1)$ integrable model possesses $U(1)$ symmetry, and common eigenstates of the transfer matrix and the Hamiltonian can be constructed as [3]

$$|\mu_1, \ldots, \mu_M\rangle = \prod_{k=1}^{M} B(\mu_1)|0\rangle_1 \otimes_s |0\rangle_2 \cdots \otimes_s |0\rangle_N, \tag{46}$$

where $\{\mu_1, \ldots, \mu_M\}$ satisfy BAEs (44) and $|0\rangle$ is the vacuum of the fermion.

## 3 $\mathfrak{gl}(1|1)$ integrable model with open boundary

### 3.1 Integrability

In this section, we consider the $\mathfrak{gl}(1|1)$ integrable model under open boundary condition. Let us introduce the $K$-matrices $K^-(u)$ and $K^+(u)$. The matrix $K^-(u)$ satisfies the graded reflection equation (RE) [43, 44]

$$R_{1,2}(u-v)K_1^-(u)R_{2,1}(u+v)K_2^-(v) = K_2^-(v)R_{1,2}(u+v)K_1^-(u)R_{2,1}(u-v), \tag{47}$$

while $K^+(u)$ satisfies the graded dual reflection equation

$$R_{1,2}(v-u)K_1^+(u)R_{2,1}(-u-v)K_2^+(v) = K_2^+(v)R_{1,2}(-u-v)K_1^+(u)R_{2,1}(v-u). \tag{48}$$

The generic solutions for the $K^\pm(u)$ are [23]

$$K^\pm(u) = \mathbb{I} + u \begin{pmatrix} a_\pm & b_\pm \mathcal{E} \\ f_\pm \mathcal{E}^\sharp & -a_\pm \end{pmatrix}, \tag{49}$$

where $a_\pm$, $b_\pm$ and $f_\pm$ are complex boundary parameters, $\mathcal{E}$ is the sole generator of complex Grassmann algebra $CG_1$, and $\mathcal{E}^\sharp$ is the adjoint of $\mathcal{E}$, i.e., $\mathcal{E}^\sharp = -i\mathcal{E}$. Further details about Grassmann numbers $\mathcal{E}$ and $\mathcal{E}^\sharp$ are provided in Appendix B.

230   We should note that for the supersymmetric $\mathfrak{gl}(1|1)$ model, the $K$-matrices must be diagonal
231 if they do not possess an additional internal space, i.e., all the elements are c-numbers. This
232 implies that in a conventional Boson-Fermion mixture, bosons cannot transform into fermions
233 upon boundary reflection. In contrast, the introduction of Grassmann numbers in Eq. (49)
234 allows for non-vanishing off-diagonal matrix elements.

235   We notice that $[K^-(u), K^+(v)] \neq 0$, which means that they cannot be diagonalized simul-
236 taneously. In this case, it is quite hard to obtain the eigenvalues via the conventional Bethe
237 ansatz methods due to the lack of a proper reference state.

238   The transfer matrix $t(u)$ is constructed as

$$t(u) = \text{str}_0\{K_0^+(u)T_0(u)K_0^-(u)\hat{T}_0(u)\}, \tag{50}$$

239 where $\hat{T}(u)$ is the reflecting monodromy matrix

$$\hat{T}_0(u) = R_{N,0}(u + \theta_N)\cdots R_{2,0}(u + \theta_2)R_{1,0}(u + \theta_1). \tag{51}$$

240   By using the graded Yang-Baxter relations (5) and reflection equations (47)-(48) repeat-
241 edly, we can prove that the transfer matrices with different spectral parameters commute with
242 each other. Therefore, $t(u)$ serves as the generating function of conserved quantities. The
243 Hamiltonian is generated from the second-order derivative of the transfer matrix [23]

$$\begin{aligned}
H &= \frac{1}{8\eta^N(1 + a_+\eta)} \left.\frac{\partial^2 t(u)}{\partial u^2}\right|_{u=0,\{\theta_j=0\}} \\
&= \sum_{j=1}^{N-1} H_{j,j+1} + \frac{\eta^{N-1}}{2}\left[a_- - 2a_- n_1 + b_-\mathcal{E}c_1 + f_-\mathcal{E}^\sharp c_1^\dagger\right] \\
&\quad + \frac{\eta^{N-1}}{2(1 + a_+\eta)}\left[a_+ - 2a_+ n_N + b_+\mathcal{E}c_N + f_+\mathcal{E}^\sharp c_N^\dagger\right].
\end{aligned} \tag{52}$$

244 The Hermiticity of Hamiltonian (52) requires $b_\pm = f_\pm^*$ and $a_\pm \in \mathbb{R}$.

## 3.2   Fusion procedure

246 The fusion approach introduced in Section 2 is also applicable to open systems. In Section 2.2,
247 we have demonstrated the fusion of the $R$-matrices and subsequently applied it to construct
248 the fused monodromy matrices given by Eq. (31).

249   The fused analogues for the reflection monodromy matrix $\hat{T}(u)$ are constructed in the same
250 way, specifically

$$\hat{T}_\alpha(u) = R_{N,\alpha}(u + \theta_N)\cdots R_{2,\alpha}(u + \theta_2)R_{1,\alpha}(u + \theta_1), \quad \alpha \in \{\bar{0}, \bar{0}', \tilde{0}, \tilde{0}'\}. \tag{53}$$

### 3.2.1   Fused $K$-matrices

252 For open systems, we should also perform the fusion procedure of the $K$-matrices using the
253 same projectors as those used for the $R$-matrices, which are introduced in Section 2.2.

254   The first-level fused $K$-matrices are

$$\begin{aligned}
K_{\bar{1}}^-(u) &= \left[[1 + (u - \tfrac{1}{2}\eta)a_-](u + \tfrac{1}{2}\eta)\right]^{-1} P_{2,1}^{(+)} K_1^-(u - \tfrac{1}{2}\eta) R_{2,1}(2u) K_2^-(u + \tfrac{1}{2}\eta) P_{1,2}^{(+)}, \\
K_{\bar{1}}^+(u) &= \left[[1 + (u + \tfrac{1}{2}\eta)a_+](u - \tfrac{1}{2}\eta)\right]^{-1} P_{1,2}^{(+)} K_2^+(u + \tfrac{1}{2}\eta) R_{1,2}(-2u) K_1^+(u - \tfrac{1}{2}\eta) P_{2,1}^{(+)}, \\
K_{\bar{1}'}^-(u) &= \left[[1 - (u + \tfrac{1}{2}\eta)a_-](u - \tfrac{1}{2}\eta)\right]^{-1} P_{2,1}^{(-)} K_1^-(u + \tfrac{1}{2}\eta) R_{2,1}(2u) K_2^-(u - \tfrac{1}{2}\eta) P_{1,2}^{(-)}, \\
K_{\bar{1}'}^+(u) &= \left[[1 - (u - \tfrac{1}{2}\eta)a_+](u + \tfrac{1}{2}\eta)\right]^{-1} P_{1,2}^{(-)} K_2^+(u - \tfrac{1}{2}\eta) R_{1,2}(-2u) K_1^+(u + \tfrac{1}{2}\eta) P_{2,1}^{(-)}.
\end{aligned} \tag{54}$$

255 The second-level fused $K$-matrices read

$$
\begin{aligned}
K_{\bar{1}}^-(u) &= \left[2[1-(u+\eta)a_-](u-\tfrac{1}{2}\eta)\right]^{-1} \mathbb{P}_{\bar{1},2}^{(-)} K_2^-(u+\eta) R_{\bar{1},2}(2u+\tfrac{1}{2}\eta) K_{\bar{1}}^-(u-\tfrac{1}{2}\eta) \mathbb{P}_{2,\bar{1}}^{(-)}, \\
K_{\bar{1}}^+(u) &= [2(1-ua_+)(u+\eta)]^{-1} \mathbb{P}_{2,\bar{1}}^{(-)} K_{\bar{1}}^+(u-\tfrac{1}{2}\eta) R_{2,\bar{1}}(-2u-\tfrac{1}{2}\eta) K_2^+(u+\eta) \mathbb{P}_{\bar{1},2}^{(-)}, \\
K_{\bar{1}'}^-(u) &= \left[2[1+(u-\eta)a_-](u+\tfrac{1}{2}\eta)\right]^{-1} \mathcal{P}_{\bar{1}',2}^{(+)} K_2^-(u-\eta) R_{\bar{1}',2}(2u-\tfrac{1}{2}\eta) K_{\bar{1}'}^-(u+\tfrac{1}{2}\eta) \mathcal{P}_{2,\bar{1}'}^{(+)}, \\
K_{\bar{1}'}^+(u) &= [2(1+ua_+)(u-\eta)]^{-1} \mathcal{P}_{2,\bar{1}'}^{(+)} K_{\bar{1}'}^+(u+\tfrac{1}{2}\eta) R_{2,\bar{1}'}(-2u+\tfrac{1}{2}\eta) K_2^+(u-\eta) \mathcal{P}_{\bar{1}',2}^{(+)}.
\end{aligned}
\tag{55}
$$

256 It should be remarked that all fused reflection matrices defined in Eqs. (54) and (55)
257 are $2 \times 2$ matrices in their respective fused spaces, and their matrix elements are operator
258 polynomials in $u$ of degree at most one. The fused $K$-matrices satisfy the following fused
259 (dual) reflection equations

$$
R_{\alpha,\beta}(u-v)K_\alpha^-(u)R_{\beta,\alpha}(u+v)K_\beta^-(v) = K_\beta^-(v)R_{\alpha,\beta}(u+v)K_\alpha^-(u)R_{\beta,\alpha}(u-v), \tag{56}
$$

$$
R_{\alpha,\beta}(v-u)K_\alpha^+(u)R_{\beta,\alpha}(-u-v)K_\beta^+(v) = K_\beta^+(v)R_{\alpha,\beta}(-u-v)K_\alpha^+(u)R_{\beta,\alpha}(v-u), \tag{57}
$$

260 where indices $\alpha, \beta$ may label either the original spaces or the projected spaces.
261 Using Eq. (29), we can finally get

$$
K_{\bar{1}}^-(u) = K_{\bar{1}'}^-(u), \quad K_{\bar{1}}^+(u) = K_{\bar{1}'}^+(u). \tag{58}
$$

262 The situation now is quite similar to the fusion of $R$-matrices described in Section 2.2. Specif-
263 ically, the $K$-matrix fusion also follows two branches that subsequently interconnect after two
264 fusion levels, as illustrated in Fig. 1 (with $R(u)$ replaced by $K^\pm(u)$).

### 3.2.2 Fused transfer matrices

266 The fused transfer matrices are defined as

$$
\begin{aligned}
t^{(1)}(u) &= \mathrm{str}_{\bar{0}}\{K_{\bar{0}}^+(u)T_{\bar{0}}(u)K_{\bar{0}}^-(u)\hat{T}_{\bar{0}}(u)\}, \\
t^{(2)}(u) &= \mathrm{str}_{\bar{0}'}\{K_{\bar{0}'}^+(u)T_{\bar{0}'}(u)K_{\bar{0}'}^-(u)\hat{T}_{\bar{0}'}(u)\}, \\
\tilde{t}^{(1)}(u) &= \mathrm{str}_{\bar{0}}\{K_{\bar{0}}^+(u)T_{\bar{0}}(u)K_{\bar{0}}^-(u)\hat{T}_{\bar{0}}(u)\}, \\
\tilde{t}^{(2)}(u) &= \mathrm{str}_{\bar{0}'}\{K_{\bar{0}'}^+(u)T_{\bar{0}'}(u)K_{\bar{0}'}^-(u)\hat{T}_{\bar{0}'}(u)\}.
\end{aligned}
\tag{59}
$$

267 From Eqs. (29), (58), and (59), it follows that the fused transfer matrices $\tilde{t}^{(1)}(u)$ and $\tilde{t}^{(2)}(u)$
268 are identical. We therefore denote them collectively as $\tilde{t}(u)$

$$
\tilde{t}(u) = \tilde{t}^{(1)}(u) = \tilde{t}^{(2)}(u). \tag{60}
$$

269 Equations (30), (56) and (57) allow us to prove that $t(u)$, $t^{(1)}(u)$, $t^{(2)}(u)$, and $\tilde{t}(u)$ are mutually
270 commutative.

### 3.3 Operator identities

272 **Operator product identities**   We introduce the function

$$
\alpha(u) = (1+ua_-)[1+(u+\eta)a_+] \prod_{j=1}^N (u+\theta_j+\eta)(u-\theta_j+\eta). \tag{61}
$$

The fused transfer matrices defined in Eq. (59) satisfy the following operator product identities

$$t(\pm\theta_j)t(\pm\theta_j + \eta) = -\frac{1}{4}\frac{\pm\theta_j(\pm\theta_j + \eta)}{(\pm\theta_j + \frac{1}{2}\eta)^2}\,\alpha(\pm\theta_j)t^{(1)}(\pm\theta_j + \tfrac{1}{2}\eta),$$

$$t(\pm\theta_j - \eta)t(\pm\theta_j) = -\frac{1}{4}\frac{\pm\theta_j(\pm\theta_j - \eta)}{(\pm\theta_j - \frac{1}{2}\eta)^2}\,\alpha(\mp\theta_j)t^{(2)}(\pm\theta_j - \tfrac{1}{2}\eta),$$

$$t^{(1)}(\pm\theta_j - \tfrac{3}{2}\eta)t(\pm\theta_j) = -\frac{\pm\theta_j(\pm\theta_j - \frac{3}{2}\eta)}{(\pm\theta_j - \frac{1}{2}\eta)(\pm\theta_j - \eta)}\,\alpha(\mp\theta_j)\tilde{t}(\pm\theta_j - \eta),$$

$$t^{(2)}(\pm\theta_j + \tfrac{3}{2}\eta)t(\pm\theta_j) = -\frac{\pm\theta_j(\pm\theta_j + \frac{3}{2}\eta)}{(\pm\theta_j + \frac{1}{2}\eta)(\pm\theta_j + \eta)}\,\alpha(\pm\theta_j)\tilde{t}(\pm\theta_j + \eta),$$

(62)

where $j = 1, \ldots, N$. A detailed proof of (62) is provided in Appendix C.

**Transfer matrices at specific points**  The properties of the $R$-matrices and $K$-matrices enable the direct evaluation of transfer matrices at specific points

$$t(0) = 0, \quad t^{(1)}(0) = 0, \quad t^{(2)}(0) = 0, \quad \tilde{t}(0) = 0, \quad t^{(1)}(-\tfrac{1}{2}\eta) = -2t(-\eta),$$
$$t^{(1)}(\tfrac{1}{2}\eta) = -2t(\eta), \quad t^{(2)}(-\tfrac{1}{2}\eta) = 2t(-\eta), \quad t^{(2)}(\tfrac{1}{2}\eta) = 2t(\eta), \quad \tilde{t}(\eta) = \tfrac{2}{3}t^{(1)}(\tfrac{3}{2}\eta).$$

(63)

**Asymptotic behavior**  Through a straightforward analysis, we obtain the following asymptotic forms of the transfer matrices $t(u)$, $t^{(1)}(u)$, $t^{(2)}(u)$ and $\tilde{t}(u)$

$$t(u)|_{u\to\infty} = 2\kappa\,u^{2N+1} \times \mathbb{I} + \cdots,$$
$$t^{(1)}(u)|_{u\to\infty} = -8\kappa\,u^{2N+1} \times \mathbb{I} + \cdots,$$
$$t^{(2)}(u)|_{u\to\infty} = 8\kappa\,u^{2N+1} \times \mathbb{I} + \cdots,$$
$$\tilde{t}(u)|_{u\to\infty} = -8\kappa\,u^{2N+1} \times \mathbb{I} + \cdots,$$

(64)

where $\kappa = a_+ + a_- + a_+a_-\eta$.

## 3.4  $T$-$Q$ relation

The transfer matrices $t(u)$, $t^{(1)}(u)$, $t^{(2)}(u)$, and $\tilde{t}(u)$ commute with each other and consequently possess common eigenstates. Let $\Lambda(u)$, $\Lambda^{(1)}(u)$, $\Lambda^{(2)}(u)$, and $\tilde{\Lambda}(u)$ denote their respective eigenvalues. Then, Eqs. (62)–(64) directly imply

$$\Lambda(\pm\theta_j)\Lambda(\pm\theta_j + \eta) = -\frac{1}{4}\frac{\pm\theta_j(\pm\theta_j + \eta)}{(\pm\theta_j + \frac{1}{2}\eta)^2}\,\alpha(\pm\theta_j)\Lambda^{(1)}(\pm\theta_j + \tfrac{1}{2}\eta),$$

$$\Lambda(\pm\theta_j - \eta)\Lambda(\pm\theta_j) = -\frac{1}{4}\frac{\pm\theta_j(\pm\theta_j - \eta)}{(\pm\theta_j - \frac{1}{2}\eta)^2}\,\alpha(\mp\theta_j)\Lambda^{(2)}(\pm\theta_j - \tfrac{1}{2}\eta),$$

$$\Lambda^{(1)}(\pm\theta_j - \tfrac{3}{2}\eta)\Lambda(\pm\theta_j) = -\frac{\pm\theta_j(\pm\theta_j - \frac{3}{2}\eta)}{(\pm\theta_j - \frac{1}{2}\eta)(\pm\theta_j - \eta)}\,\alpha(\mp\theta_j)\tilde{\Lambda}(\pm\theta_j - \eta),$$

$$\Lambda^{(2)}(\pm\theta_j + \tfrac{3}{2}\eta)\Lambda(\pm\theta_j) = -\frac{\pm\theta_j(\pm\theta_j + \frac{3}{2}\eta)}{(\pm\theta_j + \frac{1}{2}\eta)(\pm\theta_j + \eta)}\,\alpha(\pm\theta_j)\tilde{\Lambda}(\pm\theta_j + \eta),$$

(65)

where $j = 1, 2, \ldots, N$ and

$$
\Lambda(0) = 0, \quad \Lambda^{(1)}(0) = 0, \quad \Lambda^{(2)}(0) = 0, \quad \tilde{\Lambda}(0) = 0,
$$
$$
\Lambda^{(1)}(-\tfrac{1}{2}\eta) = -2\Lambda(-\eta), \qquad \Lambda^{(1)}(\tfrac{1}{2}\eta) = -2\Lambda(\eta), \tag{66}
$$
$$
\Lambda^{(2)}(-\tfrac{1}{2}\eta) = 2\Lambda(-\eta), \quad \Lambda^{(2)}(\tfrac{1}{2}\eta) = 2\Lambda(\eta), \quad \tilde{\Lambda}(\eta) = \tfrac{2}{3}\Lambda^{(1)}(\tfrac{3}{2}\eta),
$$

$$
\Lambda(u)|_{u\to\infty} = 2\kappa\, u^{2N+1} + \cdots, \qquad \Lambda^{(1)}(u)|_{u\to\infty} = -8\kappa\, u^{2N+1} + \cdots, \tag{67}
$$
$$
\Lambda^{(2)}(u)|_{u\to\infty} = 8\kappa\, u^{2N+1} + \cdots, \qquad \tilde{\Lambda}(u)|_{u\to\infty} = -8\kappa\, u^{2N+1} + \cdots.
$$

From the definitions of the transfer matrices in Eqs. (50) and (59), we know that $\Lambda(u)$, $\Lambda^{(1)}(u)$, $\Lambda^{(2)}(u)$, and $\tilde{\Lambda}(u)$ are all polynomials in $u$ of degree $2N + 2$. The $8N + 13$ equations in (65) - (67) thus provide sufficient constraints to determine these functions completely.

We can parameterize $\Lambda(u)$, $\Lambda^{(1)}(u)$, $\Lambda^{(2)}(u)$, and $\tilde{\Lambda}(u)$ by the following $T$-$Q$ relations

$$
\Lambda(u) = \frac{2u}{2u+\eta}\big[\alpha(u) - \alpha(-u-\eta)\big]\frac{Q(u-\eta)}{Q(u)},
$$
$$
\Lambda^{(1)}(u) = -\frac{4u}{u+\eta}\big[\alpha(u+\tfrac{\eta}{2}) - \alpha(-u-\tfrac{3}{2}\eta)\big]\frac{Q(u-\tfrac{3\eta}{2})}{Q(u+\tfrac{\eta}{2})},
$$
$$
\Lambda^{(2)}(u) = \frac{4u}{u+\eta}\big[\alpha(u+\tfrac{\eta}{2}) - \alpha(-u-\tfrac{3}{2}\eta)\big]\frac{Q(u-\tfrac{3\eta}{2})}{Q(u+\tfrac{\eta}{2})}, \tag{68}
$$
$$
\tilde{\Lambda}(u) = -\frac{8u}{2u+3\eta}\big[\alpha(u+\eta) - \alpha(-u-2\eta)\big]\frac{Q(u-2\eta)}{Q(u+\eta)},
$$

where

$$
Q(u) = \prod_{k=1}^{M}(u-\lambda_k)(u+\lambda_k+\eta), \quad 0 \le M \le N. \tag{69}
$$

The Bethe roots $\{\lambda_1, \ldots, \lambda_M\}$ satisfy the following BAEs

$$
\frac{\alpha(\lambda_k)}{\alpha(-\lambda_k-\eta)} = 1, \quad k = 1, \ldots M. \tag{70}
$$

The eigenvalue of the Hamiltonian (52) in terms of the Bethe roots is given by

$$
E = \frac{1}{8\eta^N(1+a_+\eta)}\left.\frac{\partial^2 \Lambda(u)}{\partial u^2}\right|_{u=0,\{\theta_j=0\}}
$$
$$
= \eta^N \sum_{k=1}^{M}\frac{1}{\lambda_k(\lambda_k+\eta)} + \frac{\eta^{N-2}}{2}\left(2N-1+a_-\eta-\frac{1}{1+a_+\eta}\right). \tag{71}
$$

Numerical results for the Bethe roots with system size $N = 3$ are presented in Table 3. We note that the eigenvalue of the Hamiltonian derived from the Bethe roots coincides with that given by the direct diagonalization of the Hamiltonian.

Since Grassmann numbers are absent from equations (65) - (67), it follows directly that the eigenvalues of the transfer matrix and the Hamiltonian are independent of them. In contrast, the eigenstates are strongly dependent on these Grassmann numbers.

We observe that the presence of boundary Grassmann numbers breaks the $U(1)$ symmetry of the system. Nevertheless, the $T$-$Q$ relations in Eq. (68) share similar structures to the ones in the periodic case (Eq. (42)). The $T$-$Q$ relation in Eq. (68) matches the earlier conjecture in Ref. [23], which was only checked numerically for small systems without an analytic proof. We address this problem by obtaining the relation analytically via the fusion approach.

The derivation of the exact spectrum of the model allows us to retrieve the Bethe state, which we will demonstrate in the following section.

Table 3: Numeric results of Bethe roots $\{\lambda_k\}$ and eigenvalues of the Hamiltonian (52) with $N = 3$, $\eta = 1$ and $a_+ = 0.5, a_- = 1.2$ and $\{\theta_j = 0\}$.

| $\lambda_1$ | $\lambda_2$ | $\lambda_3$ | $E$ |
|---|---|---|---|
| – | – | – | 2.7667 |
| $-0.5000-1.5235i$ | – | – | 2.3777 |
| $-0.5000-0.2187i$ | – | – | $-0.5911$ |
| $-0.5000-0.5565i$ | – | – | 0.9800 |
| $-0.5000-1.5235i$ | $-0.5000-0.2187i$ | – | $-0.9800$ |
| $-0.5000-1.5235i$ | $-0.5000-0.5565i$ | – | 0.5911 |
| $-0.5000-0.2187i$ | $-0.5000-0.5565i$ | – | $-2.3777$ |
| $-0.5000-1.5235i$ | $-0.5000-0.2187i$ | $-0.5000-0.5565i$ | $-2.7667$ |

## 4 Bethe state of the open $\mathfrak{gl}(1|1)$ integrable model

The Bethe-type eigenstates of integrable models with generic open boundary conditions can be constructed [21,33,34,45–47]. In this work, we apply the approach in Refs. [33,34] to retrieve the Bethe states of the open $\mathfrak{gl}(1|1)$ model. By employing two sets of gauge transformations, we obtain appropriate generators and a reference state for constructing the Bethe vectors, respectively. To verify the Bethe state, we also construct a complete basis of the Hilbert space via the separation of variables (SoV) approach [30–32].

### 4.1 Gauge transformation

For convenience, we denote the double-row monodromy matrix as

$$\mathscr{U}(u) = T(u)K^-(u)\hat{T}(u) = \begin{pmatrix} \mathscr{A}(u) & \mathscr{B}(u) \\ \mathscr{C}(u) & \mathscr{D}(u) \end{pmatrix}. \tag{72}$$

The transfer matrix $t(u)$ in Eq. (50) can be expressed as a linear combination of the elements of double-row monodromy matrix

$$t(u) = K_{11}^+(u)\mathscr{A}(u) + K_{12}^+(u)\mathscr{C}(u) - K_{21}^+(u)\mathscr{B}(u) - K_{22}^+(u)\mathscr{D}(u). \tag{73}$$

The reflection matrix $K^+(u)$ (49) can be diagonalized as follows

$$\tilde{K}^+(u) = \tilde{G}K^+(u)\tilde{G}^{-1} = \begin{pmatrix} \tilde{K}_{11}^+(u) & 0 \\ 0 & \tilde{K}_{22}^+(u) \end{pmatrix} = \begin{pmatrix} 1 + ua_+ & 0 \\ 0 & 1 - ua_+ \end{pmatrix},$$

where the gauge transformation matrix $\tilde{G}$ and its reverse $\tilde{G}^{-1}$ are

$$\tilde{G} = \frac{1}{2a_+}\begin{pmatrix} 2a_+ & b_+\mathcal{E} \\ -f_+\mathcal{E}^\sharp & 2a_+ \end{pmatrix}, \qquad \tilde{G}^{-1} = \frac{1}{2a_+}\begin{pmatrix} 2a_+ & -b_+\mathcal{E} \\ f_+\mathcal{E}^\sharp & 2a_+ \end{pmatrix}. \tag{74}$$

By applying the same gauge transformation to the $R$-matrices and $K^-(u)$, we arrive at

$$t(u) = \text{str}_0\{\tilde{K}_0^+(u)\tilde{\mathscr{U}}(u)\} = \tilde{K}_{11}^+(u)\tilde{\mathscr{A}}(u) - \tilde{K}_{22}^+(u)\tilde{\mathscr{D}}(u), \tag{75}$$

where

$$\tilde{\mathcal{U}}(u) = \tilde{G}T(u)K^-(u)\hat{T}(u)\tilde{G}^{-1} = \tilde{G}T(u)\tilde{G}^{-1}\tilde{K}^-(u)\tilde{G}\hat{T}(u)\tilde{G}^{-1} = \begin{pmatrix} \tilde{\mathcal{A}}(u) & \tilde{\mathcal{B}}(u) \\ \tilde{\mathcal{C}}(u) & \tilde{\mathcal{D}}(u) \end{pmatrix}, \qquad (76)$$

and $\tilde{K}^-(u)$ is defined as

$$\begin{aligned}
\tilde{K}^-(u) &= \tilde{G}K^-(u)\tilde{G}^{-1} = \begin{pmatrix} \tilde{K}_{11}^-(u) & \tilde{K}_{12}^-(u) \\ \tilde{K}_{21}^-(u) & \tilde{K}_{22}^-(u) \end{pmatrix} \\
&= \frac{1}{a_+} \begin{pmatrix} a_+(1 + ua_-) & (a_+b_- - b_+a_-)u\mathcal{E} \\ (a_+f_- - a_-f_+)u\mathcal{E}^\sharp & a_+(1 - ua_-) \end{pmatrix}.
\end{aligned} \qquad (77)$$

The entries of $\mathcal{U}(u)$ and $\tilde{\mathcal{U}}(u)$ satisfy the following relations

$$\begin{aligned}
\tilde{\mathcal{A}}(u) &= \mathcal{A}(u) - \frac{f_+}{2a_+}\mathcal{E}^\sharp \mathcal{B}(u) + \frac{b_+}{2a_+}\mathcal{E}\mathcal{C}(u), \\
\tilde{\mathcal{B}}(u) &= -\frac{b_+}{2a_+}\mathcal{E}\Big[\mathcal{A}(u) - \mathcal{D}(u)\Big] + \mathcal{B}(u), \\
\tilde{\mathcal{C}}(u) &= -\frac{f_+}{2a_+}\mathcal{E}^\sharp\Big[\mathcal{A}(u) - \mathcal{D}(u)\Big] + \mathcal{C}(u), \\
\tilde{\mathcal{D}}(u) &= -\frac{f_+}{2a_+}\mathcal{E}^\sharp \mathcal{B}(u) + \frac{b_+}{2a_+}\mathcal{E}\mathcal{C}(u) + \mathcal{D}(u).
\end{aligned} \qquad (78)$$

It is important to note that Grassmann numbers commute with the diagonal elements, but anti-commute with the off-diagonal elements, of the double-row monodromy matrix—a property that holds for both its original and gauge-transformed versions.

The commutation relations among $\tilde{\mathcal{A}}(u), \tilde{\mathcal{B}}(u), \tilde{\mathcal{C}}(u), \tilde{\mathcal{D}}(u)$ are the same as those among the untransformed operators. A number of useful specific relations are provided in Appendix D.

## 4.2 SoV Basis

To begin, we rewrite the $R$-matrices $R_{0,j}(u), R_{j,0}(u)$ in the auxiliary space $V_0$

$$R_{0,j}(u) = R_{j,0}(u) = \begin{pmatrix} u + \eta\,\bar{n}_j & \eta\,c_j^\dagger \\ \eta\,c_j & u - \eta\,n_j \end{pmatrix}, \qquad (79)$$

where $c_j$, $c_j^\dagger$ and $n_k$ denote the fermionic annihilation, creation, and particle number operators, respectively. By applying the gauge transformation $\tilde{G}$ to the Lax operator, we obtain

$$\tilde{R}_{0,j}(u) = \tilde{G}_0 R_{0,j}(u)\tilde{G}_0^{-1} = \begin{pmatrix} u + \eta\,\tilde{\bar{n}}_j & \eta\,\tilde{c}_j^\dagger \\ \eta\,\tilde{c}_j & u - \eta\,\tilde{n}_j \end{pmatrix}, \qquad (80)$$

where

$$\tilde{\bar{n}}_j = 1 - n_j + \rho_1 c_j - \rho_2 c_j^\dagger, \quad \tilde{c}_j^\dagger = c_j^\dagger - \rho_1, \quad \tilde{c}_j = c_j - \rho_2, \quad \tilde{n}_j = n_j - \rho_1 c_j + \rho_2 c_j^\dagger, \qquad (81)$$

and

$$\rho_1 = \frac{b_+\mathcal{E}}{2a_+}, \qquad \rho_2 = \frac{f_+\mathcal{E}^\sharp}{2a_+}. \qquad (82)$$

Let us introduce the following local state on site $n$

$$|\tilde{0}\rangle_n = |0\rangle_n - \rho_2|1\rangle_n, \quad n = 1, \cdots, N, \qquad (83)$$

335    which satisfies

$$\left[\tilde{R}_{0,j}(u)\right]_{2,1}|\tilde{0}\rangle_n = 0, \quad \left[\tilde{R}_{0,j}(u)\right]_{1,1}|\tilde{0}\rangle_n = (u+\eta)|\tilde{0}\rangle_n, \quad \left[\tilde{R}_{0,j}(u)\right]_{2,2}|\tilde{0}\rangle_n = u|\tilde{0}\rangle_n. \tag{84}$$

336    Analogously, the following local bra vector can also be constructed

$$_n\langle\tilde{0}| = {}_n\langle 0| - {}_n\langle 1|\rho_1, \quad n = 1,\cdots,N, \tag{85}$$

337    which satisfies

$$_n\langle\tilde{0}|\left[\tilde{R}_{0,j}(u)\right]_{1,2} = 0, \quad {}_n\langle\tilde{0}|\left[\tilde{R}_{0,j}(u)\right]_{1,1} = {}_n\langle\tilde{0}|(u+\eta), \quad {}_n\langle\tilde{0}|\left[\tilde{R}_{0,j}(u)\right]_{2,2} = {}_n\langle\tilde{0}|u. \tag{86}$$

338    We then introduce two global product states

$$|\omega_0\rangle = |\tilde{0}\rangle_1 \otimes_s |\tilde{0}\rangle_2 \cdots \otimes_s |\tilde{0}\rangle_N, \qquad \langle\omega_0| = {}_1\langle\tilde{0}| \otimes_s {}_2\langle\tilde{0}| \cdots \otimes_s {}_N\langle\tilde{0}|. \tag{87}$$

339    From the definition of the gauged double-row monodromy matrix, it can be shown that $|\omega_0\rangle$
340    and $\langle\omega_0|$ are eigenstates of $\tilde{\mathscr{C}}(u)$ and $\tilde{\mathscr{B}}(u)$, respectively

$$\tilde{\mathscr{C}}(u)|\omega_0\rangle = \tilde{K}_{21}^-(u)w_-(u)w_+(u+\eta)|\omega_0\rangle, \tag{88}$$

$$\langle\omega_0|\tilde{\mathscr{B}}(u) = \langle\omega_1|\tilde{K}_{12}^-(u)w_-(u+\eta)w_+(u), \tag{89}$$

341    where

$$w_\pm(u) = \prod_{j=1}^{N}(u \pm \theta_j). \tag{90}$$

342    Let's construct the SoV vectors

$$|p_1,\ldots,p_n\rangle = \tilde{\mathscr{A}}(\theta_{p_1})\ldots\tilde{\mathscr{A}}(\theta_{p_n})|\omega_0\rangle, \tag{91}$$

343    where $p_j \in \{1,\ldots,N\}$, $p_1 < p_2 < \cdots < p_n$. With the help of the following identity

$$\tilde{\mathscr{C}}(\theta_j)|\omega_0\rangle = 0, \tag{92}$$

344    and Eqs. (D.1), (D.5), we can prove that the vectors defined in Eqs. (91) are all the eigenstates
345    of $\tilde{\mathscr{C}}(u)$

$$\tilde{\mathscr{C}}(u)|p_1,\ldots,p_n\rangle = h(u,\{p_1,\ldots,p_n\})|p_1,\ldots,p_n\rangle, \tag{93}$$

346    with the corresponding eigenvalues being

$$h(u,\{p_1,\ldots,p_n\}) = \tilde{K}_{21}^-(u)w_-(u)w_+(u+\eta)\prod_{l=1}^{N}\frac{(u+\theta_{p_l})(u-\theta_{p_l}+\eta)}{(u-\theta_{p_l})(u+\theta_{p_l}+\eta)}. \tag{94}$$

We see that the vector $|p_1,\ldots,p_n\rangle$ does not depend on the order of $\tilde{\mathscr{A}}(\theta_{p_j})$, i.e.,

$$|\ldots,p_j,\ldots,p_k,\ldots\rangle = |\ldots,p_k,\ldots,p_j,\ldots\rangle.$$

347    Furthermore, vectors $\{|p_1,\ldots,p_n\rangle\}$ with distinct configurations $\{p_1,\ldots,p_n\}$ are mutually or-
348    thogonal due to the difference in their corresponding spectra. As the total number of the SoV
349    vectors in (93) equals the Hilbert space dimension, they form a complete basis.
350         Similarly, we can construct another set of Sov basis of the Hilbert

$$\langle p_1,\ldots,p_n| = \langle\omega_0|\tilde{\mathscr{A}}(-\theta_{p_1})\ldots\tilde{\mathscr{A}}(-\theta_{p_n}), \tag{95}$$

351    where $p_j \in \{1,\ldots,N\}$, $p_1 < p_2 < \cdots < p_n$. It can be proved that the vectors in Eq. (95) all are
352    eigenstates of $\tilde{\mathscr{B}}(u)$.

### 4.3 The Scalar Product $\langle\Psi|p_1,\ldots,p_n\rangle$

We introduce the scalar product

$$F_n(p_1,\ldots,p_n) = \langle\Psi|p_1,\ldots,p_n\rangle, \tag{96}$$

where $\langle\Psi|$ is a common eigenstate of the transfer matrix $t(u)$. By inserting an operator $t(\theta_{p_{n+1}})$ between the bra vector $\langle\Psi|$ and the ket vector $|p_1,\ldots,p_n\rangle$, and alternately acting it to the left and to the right, we obtain the following relation

$$\Lambda(\theta_{p_{n+1}})F_n(p_1,\ldots,p_n)$$
$$= \tilde{K}_{11}^+(\theta_{p_{n+1}})F_{n+1}(p_1,\ldots,p_n,p_{n+1}) - \tilde{K}_{22}^+(\theta_{p_{n+1}})\langle\Psi|\tilde{\mathscr{D}}(\theta_{p_{n+1}})\prod_{l=1}^{n}\tilde{\mathscr{A}}(\theta_{p_l})|\omega_0\rangle. \tag{97}$$

Introduce a useful identity

$$\tilde{\mathscr{D}}(\theta_k)|\omega_0\rangle = \frac{\eta}{2\theta_k + \eta}\tilde{\mathscr{A}}(\theta_k)|\omega_0\rangle, \quad k = 1,\ldots,N, \tag{98}$$

The commutation relations (D.5), together with Eqs. (93), (94) and (98) lead to the following identity

$$\tilde{\mathscr{D}}(\theta_{p_{n+1}})\prod_{l=1}^{n}\tilde{\mathscr{A}}(\theta_{p_l})|\omega_0\rangle = \prod_{l=1}^{n}\tilde{\mathscr{A}}(\theta_{p_l})\tilde{\mathscr{D}}(\theta_{p_{n+1}})|\omega_0\rangle$$
$$= \frac{\eta}{2\theta_{p_{n+1}} + \eta}\prod_{l=1}^{n}\tilde{\mathscr{A}}(\theta_{p_l})\tilde{\mathscr{A}}(\theta_{p_{n+1}})|\omega_0\rangle = \frac{\eta}{2\theta_{p_{n+1}} + \eta}|p_1,\ldots,p_n,p_{n+1}\rangle. \tag{99}$$

Therefore, we obtain

$$\Lambda(\theta_{p_{n+1}})F_n(p_1,\ldots,p_n) = \frac{(2\theta_{p_{n+1}} + \eta)\tilde{K}_{11}^+(\theta_{p_{n+1}}) - \eta\tilde{K}_{22}^+(\theta_{p_{n+1}})}{2\theta_{p_{n+1}} + \eta}F_{n+1}(p_1,\ldots,p_{n+1}), \tag{100}$$

which allows us to get the expression of $\{F_n(p_1,\ldots,p_n)\}$

$$F_n(p_1,\ldots,p_n) = \prod_{l=1}^{n}\frac{(2\theta_{p_l} + \eta)\Lambda(\theta_{p_l})}{(2\theta_{p_l} + \eta)\tilde{K}_{11}^+(\theta_{p_l}) - \eta\tilde{K}_{22}^+(\theta_{p_l})}F_0, \tag{101}$$

where $F_0 = \langle\Psi|\omega_0\rangle$ is an overall factor. Substituting the explicit expression of the eigenvalue $\Lambda(u)$ given by $T$-$Q$ relation (68), we further derive

$$F_n(p_1,\ldots,p_n) = \prod_{l=1}^{n}(1 + \theta_{p_l}a_-)w_-(\theta_{p_l} + \eta)w_+(\theta_{p_l} + \eta)\frac{Q(\theta_{p_l} - \eta)}{Q(\theta_{p_l})}F_0, \tag{102}$$

where $Q(u)$ is defined in Eq. (69). Since the SoV basis is complete, the set $\{F_n(p_1,\ldots,p_n)\}$ can completely determine the form of Bethe state $\langle\Psi|$.

### 4.4 Bethe state

Introduce another gauge transformation

$$\bar{G} = \tilde{G}\big|_{\{a_+,b_+,f_+\}\rightarrow\{a_-,b_-,f_-\}}, \tag{103}$$

so that $K^-(u)$ becomes diagonal under this transformation

$$\bar{K}^-(u) = \bar{G}K_-(u)\bar{G}^{-1} = \begin{pmatrix} \bar{K}_{11}^-(u) & 0 \\ 0 & \bar{K}_{22}^-(u) \end{pmatrix} = \begin{pmatrix} 1 + ua_- & 0 \\ 0 & 1 - ua_- \end{pmatrix}. \tag{104}$$

Applying the same gauge transformation to the double-row monodromy matrix yields

$$\bar{\mathscr{U}}(u) = \bar{G}\mathscr{U}G^{-1} = \begin{pmatrix} \bar{\mathscr{A}}(u) & \bar{\mathscr{B}}(u) \\ \bar{\mathscr{C}}(u) & \bar{\mathscr{D}}(u) \end{pmatrix}. \tag{105}$$

Define the following global vectors

$$|\bar{\omega}_0\rangle = |\omega_0\rangle_{\{a_+,b_+,f_+\}\to\{a_-,b_-,f_-\}}, \quad \langle\bar{\omega}_0| = \langle\omega_0|_{\{a_+,b_+,f_+\}\to\{a_-,b_-,f_-\}}. \tag{106}$$

The state $\langle\bar{\omega}_0|$ in (106) satisfies

$$\langle\bar{\omega}_0|\bar{\mathscr{B}}(u) = 0, \quad \langle\bar{\omega}_0|\bar{\mathscr{A}}(u) = \bar{K}_{11}^-(u)w_-(u+\eta)w_+(u+\eta)\langle\bar{\omega}_0|. \tag{107}$$

The aforementioned equation (107) together with two other identities

$$\langle\bar{\omega}_0|\tilde{\mathscr{C}}(\theta_k)|p_1,\ldots,p_n\rangle = 0, \quad k \notin \{p_1,\ldots,p_n\}, \tag{108}$$

$$\tilde{\mathscr{A}}(u) = \left(\frac{f_-}{2a_-} - \frac{f_+}{2a_+}\right)\mathcal{E}^\sharp\bar{\mathscr{B}}(u) + \left(\frac{b_+}{2a_+} - \frac{b_-}{2a_-}\right)\mathcal{E}\tilde{\mathscr{C}}(u) + \bar{\mathscr{A}}(u), \tag{109}$$

allow us to get

$$\langle\bar{\omega}_0|p_1,\ldots,p_n,p_{n+1}\rangle = \langle\bar{\omega}_0|\tilde{\mathscr{A}}(\theta_{n+1})|p_1,\ldots,p_n\rangle$$
$$= \langle\bar{\omega}_0|\bar{\mathscr{A}}(\theta_{n+1})|p_1,\ldots,p_n\rangle$$
$$= \bar{K}_{11}^-(\theta_{p_{n+1}})w_+(\theta_{p_{n+1}}+\eta)w_-(\theta_{p_{n+1}}+\eta)\langle\bar{\omega}_0|p_1,\ldots,p_n\rangle. \tag{110}$$

Furthermore, we can derive the expression of the overlap $\langle\bar{\omega}_0|p_1,\ldots,p_n\rangle$ from the recursive relation (110)

$$\langle\bar{\omega}_0|p_1,\ldots,p_n\rangle = \prod_{k=1}^{n}\bar{K}_{11}^-(\theta_{p_k})w_+(\theta_{p_k}+\eta)w_-(\theta_{p_k}+\eta)\langle\bar{\omega}_0|\omega_0\rangle. \tag{111}$$

**Bethe state** The left Bethe state can be parameterized as

$$\langle\lambda_1,\ldots,\lambda_N| = \langle\bar{\omega}_0|\prod_{l=1}^{M}\tilde{\mathscr{C}}(\lambda_l), \tag{112}$$

where $\{\lambda_1,\ldots,\lambda_N\}$ are the Bethe roots satisfying BAEs (70), the generator $\tilde{\mathscr{C}}(u)$ and the reference state $\langle\bar{\omega}_0|$ are defined in Eqs. (76) and (106) respectively.

The proof of our Bethe state is straightforward. A combination of Eqs. (93), (94), and (111) yields

$$\langle\lambda_1,\ldots,\lambda_N|p_1,\ldots,p_n\rangle = \prod_{l=1}^{n}(1+\theta_{p_l}a_-)w_-(\theta_{p_l}+\eta)w_+(\theta_{p_l}+\eta)\frac{Q(\theta_{p_l}-\eta)}{Q(\theta_{p_l})}$$
$$\times \prod_{k=1}^{M}\bar{K}_{21}^-(\lambda_k)w_-(\lambda_k)w_+(\lambda_k+\eta)\langle\bar{\omega}_0|\omega_0\rangle. \tag{113}$$

The factor on the second line of Eq. (113) is a normalization factor. By comparing Eqs. (102) and (113), we can conclude that $\langle\lambda_1,\ldots,\lambda_N|$ is an eigenstate of the transfer matrix.

384      Analogously, the right Bethe state can also be constructed

$$|\lambda_1, \ldots, \lambda_N\rangle = \prod_{l=1}^{M} \tilde{\mathscr{B}}(\lambda_l)|\bar{\omega}_0\rangle. \tag{114}$$

385      It should be remarked that the generation operators, the Bethe roots and the reference
386 states in Eqs. (112) and Eqs. (114) all have well-defined homogeneous limits of $\{\theta_j \to 0\}$.
387      Under the condition $a_- f_+ = f_- a_+$, the state $|\bar{\omega}_0\rangle$ reduces to $|\omega_0\rangle$, and the resulting Bethe
388 state (114) coincides with the one given in Ref. [23]. In this case, we can use the gauge matrix
389 $\tilde{G}$ to simultaneously diagonalize $K^+(u)$ and triangularize $K^-(u)$ (see Eq. (77)), making the
390 conventional algebraic Bethe ansatz applicable.

## 5   Conclusion

392 The exact solution of the supersymmetric $\mathfrak{gl}(1|1)$ integrable models with both periodic and
393 generic non-diagonal open boundary conditions is presented in this paper. Using the fusion
394 procedure, we construct a hierarchy of fused transfer matrices, from which a closed set of
395 operator identities is derived. These identities yield the energy spectrum of the model, includ-
396 ing the $T$-$Q$ relation and the corresponding Bethe ansatz equations. With the exact spectrum
397 obtained, we then construct the corresponding Bethe states, notably for the open chain with
398 generic non-diagonal boundary conditions.
399      The method developed in this work can be applied to other quantum integrable models
400 associated with Lie superalgebra. In particular, it extends straightforwardly to the $U_q(\mathfrak{gl}(1|1))$
401 quantum algebra, for which the $R$–matrix and the reflection $K$–matrices retain the same graded
402 structure as those of the undeformed $\mathfrak{gl}(1|1)$ superalgebra [48]. In a parallel investigation of
403 the quantum integrable model associated with the Lie superalgebra $\mathfrak{gl}(2|2)$, we have succeeded
404 in establishing virtually all of the operator identities. For higher rank cases, the fusion proce-
405 dure involves additional levels and branching structures.

## Acknowledgments

407 We thank Prof. Wen-Li Yang for valuable discussions. Financial supports from the National Key
408 R&D Program of China (Grant No. 2021YFA1402104), National Natural Science Foundation
409 of China (Grant Nos. 12105221, 12247103, 12074410, 12047502, 12434006, 12575007),
410 Shaanxi Fundamental Science Research Project for Mathematics and Physics (Grant Nos. 22JSZ005),
411 Scientific Research Program Funded by Shaanxi Provincial Education Department (Grant No.
412 21JK0946), Beijing National Laboratory for Condensed Matter Physics (Grant No. 202162100001),
413 and Double First-Class University Construction Project of Northwest University are acknowl-
414 edged.

## A   The second fusion branch

416 Let us introduce the second fusion branch of $R$-matrix in Section 2.2.2 detailedly. When
417 $u = -\eta$, the $R$-matrix in (1) becomes

$$R_{1,2}(-\eta) = -2\eta P_{1,2}^{(-)} = -2\eta(1 - P_{1,2}^{(+)}), \tag{A.1}$$

418   where $P_{1,2}^{(-)}$ is a 2-dimensional supersymmetric projector with the following form

$$P_{1,2}^{(-)} = \sum_{i=1}^{2} |\bar{\psi}_i\rangle\langle\bar{\psi}_i|, \qquad P_{1,2}^{(-)} = P_{2,1}^{(-)}, \tag{A.2}$$

$$|\bar{\psi}_1\rangle = \frac{1}{\sqrt{2}}(|1,2\rangle - |2,1\rangle), \quad |\bar{\psi}_2\rangle = |2,2\rangle. \tag{A.3}$$

419   The corresponding parities are

$$p(\bar{\psi}_1) = 1, \quad p(\bar{\psi}_2) = 0.$$

420   The operator $P_{1,2}^{(-)}$ projects the 4-dimensional product space $V_1 \otimes_s V_2$ into a new 2-dimensional
421   space spanned by $\{|\bar{\psi}_i\rangle | i = 1, 2\}$.
422      By fusing the $R$-matrix with this projector $P_{1,2}^{(-)}$, we can obtain the specific form of $R_{\bar{1}',n}(u)$
423   defined in (24), which is

$$R_{\bar{1}',n}(u) = \begin{pmatrix} u + \frac{3}{2}\eta & & & \\ & u - \frac{1}{2}\eta & -\sqrt{2}\eta & \\ & -\sqrt{2}\eta & u + \frac{1}{2}\eta & \\ & & & u - \frac{3}{2}\eta \end{pmatrix}. \tag{A.4}$$

424      At the point of $u = \frac{3}{2}\eta$, the fused $R$-matrix $R_{\bar{1}',2}(u)$ in (24) degenerates into

$$R_{\bar{1}',2}(\tfrac{3}{2}\eta) = 3\eta \mathcal{P}_{\bar{1}',2}^{(+)}, \tag{A.5}$$

425   where $\mathcal{P}_{\bar{1}',2}^{(+)}$ is a 2-dimensional supersymmetric projector with the form of

$$\mathcal{P}_{\bar{1}',2}^{(+)} = \sum_{i=1}^{2} |\tilde{\phi}_i\rangle\langle\tilde{\phi}_i|, \tag{A.6}$$

426   and the corresponding vectors are

$$|\tilde{\phi}_1\rangle = |\bar{\psi}_1\rangle \otimes_s |1\rangle, \quad |\tilde{\phi}_2\rangle = \frac{1}{\sqrt{3}}(\sqrt{2}|\bar{\psi}_2\rangle \otimes_s |1\rangle - |\bar{\psi}_1\rangle \otimes_s |2\rangle). \tag{A.7}$$

427   Here, the $|\bar{\psi}_1\rangle$ and $|\bar{\psi}_2\rangle$ are given in Eq. (A.3). The parities read

$$p(\tilde{\phi}_1) = 1, \quad p(\tilde{\phi}_2) = 0.$$

428      Similarly, we can get the specific form of the $R_{\bar{1}',n}(u)$ given in Eq. (27)

$$R_{\bar{1}',n}(u) = \begin{pmatrix} u + 2\eta & & & \\ & u - \eta & -\sqrt{3}\eta & \\ & -\sqrt{3}\eta & u + \eta & \\ & & & u - 2\eta \end{pmatrix}. \tag{A.8}$$

429   From Eqs. (22) and (A.8), we can easily see that $R_{\bar{1},2}(u)$ given by (20) and $R_{\bar{1}',2}(u)$ given by
430   (27) are the same, i.e., Eq. (29).

## B  Grassmann Numbers

Grassmann numbers are the anticommuting algebraic variables that play a central role in supersymmetric models and integrable systems with $\mathbb{Z}_2$ grading. The Grassmann algebra $CG_N$ is generated by $N$ generators $\mathcal{E}_1, \mathcal{E}_2, \cdots, \mathcal{E}_N$, where the generators satisfy the nilpotency condition

$$\mathcal{E}_i^2 = 0, \tag{B.1}$$

and the anticommutation relations

$$\mathcal{E}_i \mathcal{E}_j = -\mathcal{E}_j \mathcal{E}_i. \tag{B.2}$$

## C  Proof of Eq. (62)

We know that the reflecting monodromy matrix $\hat{T}(u)$ in Eq. (51) and its fused analogues satisfy the graded RTT relations

$$R_{\alpha,\beta}(u-v)\hat{T}_\alpha(u)\hat{T}_\beta(v) = \hat{T}_\beta(v)\hat{T}_\alpha(u)R_{\alpha,\beta}(u-v), \tag{C.1}$$

where the indices $\alpha, \beta$ may label either the original spaces or the projected spaces.

Because the (fused) $R$-matrices collapse to projectors at certain special values of the spectral parameter, the (fused) monodromy matrices $\hat{T}_\alpha(u)$ satisfy the following relations

$$P_{1,2}^{(+)}\hat{T}_1(u)\hat{T}_2(u+\eta)P_{1,2}^{(+)} = \prod_{l=1}^{N}(u+\theta_l+\eta)\hat{T}_{\bar{1}}(u+\tfrac{1}{2}\eta),$$

$$P_{1,2}^{(-)}\hat{T}_1(u)\hat{T}_2(u-\eta)P_{1,2}^{(-)} = \prod_{l=1}^{N}(u+\theta_l-\eta)\hat{T}_{\bar{1}'}(u-\tfrac{1}{2}\eta),$$

$$\mathbb{P}_{2,\bar{1}}^{(-)}\hat{T}_2(u+\eta)\hat{T}_{\bar{1}}(u-\tfrac{1}{2}\eta)\mathbb{P}_{2,\bar{1}}^{(-)} = \prod_{l=1}^{N}(u+\theta_l)\hat{T}_{\tilde{1}}(u), \tag{C.2}$$

$$\mathcal{P}_{2,\bar{1}'}^{(+)}\hat{T}_2(u-\eta)\hat{T}_{\bar{1}'}(u+\tfrac{1}{2}\eta)\mathcal{P}_{2,\bar{1}'}^{(+)} = \prod_{l=1}^{N}(u+\theta_l)\hat{T}_{\tilde{1}'}(u),$$

where the projectors $P_{1,2}^{(+)}, \mathbb{P}_{2,\bar{1}}^{(-)}, P_{1,2}^{(-)}$ and $\mathcal{P}_{2,\bar{1}'}^{(+)}$ are given by (12),(18), (A.2) and (A.6), respectively.

We define the degenerate point of the $R$-matrix as $\delta$, at which we have $R_{\alpha,\beta}(\delta) = P_{\alpha,\beta}^{(d)}S_{\alpha,\beta}$, where $P_{\alpha,\beta}^{(d)}$ is a $d$-dimensional projector and $S_{\alpha,\beta}$ is a constant matrix. Employing the property of the projector that $P_{\alpha,\beta}^{(d)}R_{\alpha,\beta}(\delta) = R_{\alpha,\beta}(\delta)$, the RTT relations (7) and (32) at the degenerate point give

$$T_\alpha(u)T_\beta(u+\delta)P_{\beta,\alpha}^{(d)} = P_{\beta,\alpha}^{(d)}T_\alpha(u)T_\beta(u+\delta)P_{\beta,\alpha}^{(d)}. \tag{C.3}$$

Similarly, from the graded RTT relations (C.1), we have

$$\hat{T}_\alpha(u)\hat{T}_\beta(u+\eta)P_{\alpha,\beta}^{(d)} = P_{\alpha,\beta}^{(d)}\hat{T}_\alpha(u)\hat{T}_\beta(u+\eta)P_{\alpha,\beta}^{(d)}, \tag{C.4}$$

Using the properties of projector, one can derive the following identities from Eq. (C.2)

$$\hat{T}_1(-\theta_j)\hat{T}_2(-\theta_j+\eta) = P_{1,2}^{(+)}\hat{T}_1(-\theta_j)\hat{T}_2(-\theta_j+\eta),$$

$$\hat{T}_1(-\theta_j)\hat{T}_2(-\theta_j-\eta) = P_{1,2}^{(-)}\hat{T}_1(-\theta_j)\hat{T}_2(-\theta_j-\eta),$$

$$\hat{T}_2(-\theta_j)\hat{T}_{\bar{1}}(-\theta_j-\tfrac{3}{2}\eta) = \mathbb{P}_{2,\bar{1}}^{(-)}\hat{T}_2(-\theta_j)\hat{T}_{\bar{1}}(-\theta_j-\tfrac{3}{2}\eta), \tag{C.5}$$

$$\hat{T}_2(-\theta_j)\hat{T}_{\bar{1}'}(-\theta_j+\tfrac{3}{2}\eta) = \mathcal{P}_{2,\bar{1}'}^{(+)}\hat{T}_2(-\theta_j)\hat{T}_{\bar{1}'}(-\theta_j+\tfrac{3}{2}\eta),$$

where $j = 1, \ldots, N$.

We can combine Eq. (36) for the monodromy matrices $T_\alpha(u)$ and Eq. (C.5) for the reflecting monodromy matrices $\hat{T}_\alpha(u)$ and finally get the following equations

$$t(u)t(u+\eta) = [\rho_2(2u+\eta)]^{-1}\mathrm{str}_{1,2}\{K_2^+(u+\eta)R_{1,2}(-2u-\eta)K_1^+(u)T_1(u)T_2(u+\eta)$$
$$\times K_1^-(u)R_{2,1}(2u+\eta)K_2^-(u+\eta)\hat{T}_1(u)\hat{T}_2(u+\eta)\},$$
(C.6)

$$t^{(1)}(u-\tfrac{1}{2}\eta)t(u+\eta) = [\rho_3(2u+\tfrac{1}{2}\eta)]^{-1}\mathrm{str}_{\bar{1},2}\{K_{\bar{1}}^+(u-\tfrac{1}{2}\eta)R_{2,\bar{1}}(-2u-\tfrac{1}{2}\eta)K_2^+(u+\eta)$$
$$\times T_2(u+\eta)T_{\bar{1}}(u-\tfrac{1}{2}\eta)K_2^-(u+\eta)R_{\bar{1},2}(2u+\tfrac{1}{2}\eta)K_{\bar{1}}^-(u-\tfrac{1}{2}\eta)\hat{T}_2(u+\eta)\hat{T}_{\bar{1}}(u-\tfrac{1}{2}\eta)\},$$
(C.7)

$$t^{(2)}(u+\tfrac{1}{2}\eta)t(u-\eta) = [\rho_4(2u-\tfrac{1}{2}\eta)]^{-1}\mathrm{str}_{\bar{1}',2}\{K_{\bar{1}'}^+(u+\tfrac{1}{2}\eta)R_{2,\bar{1}'}(-2u+\tfrac{1}{2}\eta)K_2^+(u-\eta)$$
$$\times T_2(u-\eta)T_{\bar{1}'}(u+\tfrac{1}{2}\eta)K_2^-(u-\eta)R_{\bar{1}',2}(2u-\tfrac{1}{2}\eta)K_{\bar{1}'}^-(u+\tfrac{1}{2}\eta)\hat{T}_2(u-\eta)\hat{T}_{\bar{1}'}(u+\tfrac{1}{2}\eta)\}.$$
(C.8)

Substituting Eq. (38), (54)-(55) and (C.3)-(C.5) into Eq. (C.6) and letting $u = \pm\theta_j, \pm\theta_j-\eta$ respectively, we get the first two lines of Eq. (62); substituting Eq. (38), (54)-(55) and (C.3)-(C.5) into Eq. (C.7) and letting $u = \pm\theta_j-\eta$, we get the third line of Eq. (62); substituting Eq. (38), (54)-(55) and (C.3)-(C.5) into Eq. (C.8) and letting $u = \pm\theta_j+\eta$, we get the fourth line of Eq. (62).

# D   Commutation relations

Some useful commutation relations used in Section 4 are

$$\tilde{\mathscr{C}}(u)\tilde{\mathscr{A}}(v) = \frac{(u-v+\eta)(u+v)}{(u+v+\eta)(u-v)}\tilde{\mathscr{A}}(v)\tilde{\mathscr{C}}(u)$$
$$-\frac{\eta}{u+v+\eta}\Big\{\tilde{\mathscr{D}}(u)\tilde{\mathscr{C}}(v)+\frac{u+v}{u-v}\tilde{\mathscr{A}}(u)\tilde{\mathscr{C}}(v)\Big\},$$
(D.1)

$$\tilde{\mathscr{D}}(v)\tilde{\mathscr{C}}(u) = \frac{(u-v-\eta)(u+v)}{(u+v-\eta)(u-v)}\tilde{\mathscr{C}}(u)\tilde{\mathscr{D}}(v)$$
$$+\frac{\eta}{u+v-\eta}\Big\{\tilde{\mathscr{C}}(v)\tilde{\mathscr{A}}(u)+\frac{u+v}{u-v}\tilde{\mathscr{C}}(v)\tilde{\mathscr{D}}(u)\Big\},$$
(D.2)

$$\tilde{\mathscr{A}}(u)\tilde{\mathscr{A}}(v) = \tilde{\mathscr{A}}(v)\tilde{\mathscr{A}}(u)+\frac{\eta}{u+v+\eta}\Big\{\tilde{\mathscr{B}}(v)\tilde{\mathscr{C}}(u)-\tilde{\mathscr{B}}(u)\tilde{\mathscr{C}}(v)\Big\},$$
(D.3)

$$\tilde{\mathscr{D}}(u)\tilde{\mathscr{D}}(v) = \tilde{\mathscr{D}}(v)\tilde{\mathscr{D}}(u)-\frac{\eta}{u+v-\eta}\Big\{\tilde{\mathscr{C}}(v)\tilde{\mathscr{B}}(u)-\tilde{\mathscr{C}}(u)\tilde{\mathscr{B}}(v)\Big\},$$
(D.4)

$$\tilde{\mathscr{D}}(u)\tilde{\mathscr{A}}(v) = \tilde{\mathscr{A}}(v)\tilde{\mathscr{D}}(u)-\frac{\eta(u+v)}{(u-v)(u+v+\eta)}\Big\{\tilde{\mathscr{B}}(v)\tilde{\mathscr{C}}(u)-\tilde{\mathscr{B}}(u)\tilde{\mathscr{C}}(v)\Big\},$$
(D.5)

$$\tilde{\mathscr{B}}(u)\tilde{\mathscr{B}}(v) = -\frac{u-v-\eta}{u-v+\eta}\tilde{\mathscr{B}}(v)\tilde{\mathscr{B}}(u),$$
(D.6)

$$\tilde{\mathscr{C}}(u)\tilde{\mathscr{C}}(v) = -\frac{u-v+\eta}{u-v-\eta}\tilde{\mathscr{C}}(v)\tilde{\mathscr{C}}(u).$$
(D.7)

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
