# Peer review of "Fusion approach for quantum integrable system associated with the gl(1|1) Lie superalgebra"

_SciPost Physics_

## Round 1 · Referee Report · Anonymous (Referee 1) · 2025-12-12

Strengths

1- exact solution of the gl(1|1) superspin chain for various boundary conditions based on operator identities derived from the fusion hierarchy

Weaknesses

1- the results are not sufficiently discussed in the context of previous work on this problem 2- the results of the present approach are still limited to eigenvalues of the transfer matrix, the dependence of eigenstates on the off-diagonal elements of the reflection matrices appears to be still out of reach.

Report

The authors present the solution of an integrable system based on the superalgebra gl(1|1) subject to periodic and generic open boundary conditions. For both types of BCs they derive a set of operator identities satisfied by the (finite) hierarchy of fused transfer matrices of the model. The resulting functional equations for their eigenvalues yield the Bethe equations for this model.

For periodic BC the model is a free fermion model and the Bethe equations (44) describe the quantisation of single particle momenta as found by elementary methods.

Generic open boundary conditions allowing for the transformation of bosons into fermions and vice versa are described by reflection matrices with Grassmann valued off-diagonal elements.
The Bethe eqs. (68) derived in the present manuscript coincide with the ones derived in [23] for diagonal and 'quasi-diagonal' BCs (essentially one diagonal and one triangular reflection matrix) using the graded algebraic Bethe ansatz on a reference state constructed from a fermionic coherent state.

For generic off-diagonal BC only the diagonal elements of the reflection matrix enter in the transfer matrix eigenvalues and Bethe eqs. The latter appear to coincide with the ones proposed (and verified for small systems) in Ref. [23] based on a single TQ-relation (i.e. the first of the relations (67)).

In summary, the authors have rederived the Bethe equations describing the spectrum of an integrable superspin chain based on gl(1|1)-symmetric R-matrices. Their analysis is based on operator identities following from the fusion hierarchy of transfer matrices and complements the construction used in Ref. [23] where a single TQ-relation for generic BCs has been 'guessed' from the one for diagonal or quasi-diagonal ones.

Requested changes

1- The authors should add a discussion of their results in the context of those from Ref. [23] 2- If possible they should also extend their remarks at the end of Section 3 concerning the construction of a reference state and/or the application of SoV to construct eigenstates of the model.

Recommendation

Ask for minor revision

---

## Editorial Decision

in_refereeing